# VR Multiscale Geovisualization Based on UAS Multitemporal Data: The Case of Geological Monuments

**Ermioni-Eirini Papadopoulou** [1,*] , **Apostolos Papakonstantinou** [2] , **Nikoletta-Anna Kapogianni** [3] , **Nikolaos Zouros** [1] and **Nikolaos Soulakellis** [1]

1   Department of Geography, University of the Aegean, 81100 Mytilene, Greece
2   Department of Marine Sciences, University of the Aegean, 81100 Mytilene, Greece
3   Department of Informatics, University of Piraeus, 18534 Piraeus, Greece
*   Correspondence: epapa@geo.aegean.gr; Tel.: +30-22510-36428

**Abstract:** Technological progress in Virtual Reality (VR) and Unmanned Aerial Systems (UASs) offers great advantages in the field of cartography and particularly in the geovisualization of spatial data. This paper investigates the correlation between UAS flight characteristics for data acquisition and the quality of the derived maps and 3D models of geological monuments for VR geovisualization in different scales and timeframes. In this study, we develop a methodology for mapping geoheritage monuments based on different cartographic scales. Each cartographic scale results in diverse orthophotomaps and 3D models. All orthophotomaps and 3D models provide an optimal geovisualization, combining UAS and VR technologies and thus contributing to the multitemporal 3D geovisualization of geological heritage on different cartographic scales. The study area selected was a fossilite ferrous site located in Lesvos Geopark, UNESCO. The study area contains a fossil site surrounding various findings. The three distinct scales that occur are based on the object depicted: (i) the fossilite ferrous site (1:120), (ii) the fossil root system (1:20), and (iii) individual fossils (≥1:10). The methodology followed in the present research consists of three main sections: (a) scale-variant UAS data acquisition, (b) data processing and results (2D–3D maps and models), and (c) 3D geovisualization to VR integration. Each different mapping scale determines the UAS data acquisition parameters (flight pattern, camera orientation and inclination, height of flight) and defines the resolution of the 3D models to be embedded in the VR environment. Due to the intense excavation of the study area, the location was spatiotemporally monitored on the cartographic scale of 1:120. For the continuous monitoring of the study area, four different UASs were also used. Each of them was programmed to fly and acquire images with a constant ground sampling distance (GSD). The data were processed by image-based 3D modeling and computer vision algorithms from which the 3D models and orthophotomaps were created and used in the VR environment. As a result, a VR application visualizing multitemporal data of geoheritage monuments across three cartographic scales was developed.

**Keywords:** geovisualization; UAS; VR; multiscale; multitemporal; geoheritage

## 1. Introduction

Mapping techniques with remote sensing and three-dimensional (3D) earth modeling have now achieved significant progress both in terms of vehicles and sensors, as well as the methods and software used [1].

In the last decade, UASs received a lot of attention as platforms equipped with recording sensors capable of automated missions for quick 2D and 3D data production [2–5]. The advantage of UAS systems is their ability to map and monitor with high temporal and spatial resolution various phenomena on the earth's surface [6–8]. Moreover, UASs allow a quick, easy, and low-cost method of data acquisition in a number of critical situations where immediate access to 3D geo-information is crucial [7,9–11]. They can be used in high-risk

situations and inaccessible areas to monitor spatiotemporal changes and phenomena. In some cases where a large cartographic scale is demanded, UASs can be a complement to or a replacement of terrestrial acquisition [12]. The high-resolution aerial images acquired by UASs can be used not only for the generation of dense point clouds, but for texture mapping of 3D data, for orthophoto production, high-detail digital surface model creation, or 3D building modeling [13–20]. UASs decrease operational costs and reduce the risk of access in harsh environments while still keeping a high accuracy potential [12].

In the last decade, parallel to the advancements in UAS technology, the spatial resolution of satellite imagery has significantly improved. Until now, satellite data are still not sufficient for mapping and monitoring very small changes (centimetric precision). Combined with various miniaturized high-precision sensors, UASs can provide high-resolution aerial images in combination with Ground Control Points (GCPs), and Post Processing Kinematic (PPK) and Real Time Kinematic (RTK) methods, to create accurate geo-information at a low cost [21,22]. Unmanned aerial systems (UASs), consisting of a UAS and a sensor, provide digital images with spatial and temporal resolutions capable of overcoming some of the limitations of spatial data acquisition using satellites and airplanes. The increase in the flight capabilities and agility of UASs, as well as in the endurance and the variation of the onboard sensors and tools available, can be used for various monitoring tasks for a plethora of spatiotemporal phenomena and environmental parameters [23–25]. Several recent publications have described methods and techniques that measure spatiotemporal changes using UASs [5,15,26,27]. UASs are a viable option for collecting remote sensing data for a wide range of practical applications, including scientific, agricultural, and environmental applications [28–31].

New technologies in remote sensing that emerged in the 21st century and the advent of UAS in data acquisition changed the mapping process and reshaped visualization products. UAS aerial data offer a unique opportunity to measure, analyze, quantify, map, and explore phenomena at high temporal frequencies and at very high ground resolution [32,33]. High-resolution and scale-variant UAS data contribute significantly to the cartographic and visualization process of spatiotemporal geo-information [5]. Thus, new visualizations can be implemented to support the identification and visualization of new patterns, potential relationships related to the spatiotemporal trends of various phenomena, and geo-information extraction.

Geovisualization, as a subcategory of scientific visualization in the field of computer technology, is the visualization of geospatial data, where spatial patterns and relationships in complex data are interpreted using visual representations [34,35]. Interactive multimedia technology has contributed substantially to the development of geovisualization tools in cartography, with multimedia cartography being a new and efficient way of accessing and delivering geospatial information, both for professionals in the field and the public who now utilize maps on a daily basis [36]. This method extends beyond the context of traditional cartography, requiring further investigation on how it can be utilized in the development of already available and continuously growing geospatial databases [37–40]. Despite technological advances and the variety of 3D visualization applications available, where spatial data is conveyed through different scales and from global to local levels, there is still unknown ground on a conceptual basis and in how they affect human perception [41].

In the era of digital information technologies, 3D modeling and computer graphics techniques apply to the development of virtual models for computer simulation, artificial intelligence (AI), big data analytics, etc., and to various applications in virtual reality (VR) [42]. Virtual Reality (VR) is the technology that provides an almost real and believable experience in a synthetic or virtual way. The goal of immersive VR is to completely immerse the user inside the computer-generated world, giving the impression that the user has "entered" the synthetic world. With the right level of immersion, VR can support a wide range of uses, including:(i) training and education [43], (ii) customer experience [44,45], (iii) entertainment [46], (iv) travel and tourism [47,48], and nowadays (v) geovisualization [26,49].

The number of contemporary visualization techniques applied to geographic data is rapidly increasing [50–53]. VR techniques challenge the means of geovisualization, as their potential applications in cartography and geoinformatics have not been thoroughly investigated [54–57]. In the present studies, references to VR applications present implementations and approaches to the visualization of geographic data [58–61]. However, the utilization of 2D and 3D maps in VR has been assessed only to a limited extent [62–65]. Although photogrammetric 3D models derived from images acquired from UAS are employed in VR, they do not always conform to established cartographic principles. In the recent literature, geovisualizations in VR are carried out using 3D terrain models and 3D point clouds [66,67]. Terrain models and VR point clouds concern smaller cartographic scales such as entire drainage basins or large geographical areas [68,69]. In the existing studies, a comparison is made between different recording sensors such as high-resolution cameras and LiDAR, only for the same area but not for various scales [66]. Based on the research to date regarding geovisualizations in a VR environment, it is observed that neither is the concept of cartographic scale considered, nor is a flight mode (flight height, camera angle, etc.) proposed for the collection of high-resolution images for the accurate and precise creation of 3D models of geographical areas. In addition, a way of visualizing spatiotemporal data and monitoring dynamic phenomena in virtual reality is not present in prior research. Thus, to cover the above gaps, this work focuses on proposing a way to collect data according to the cartographic scale to produce 3D models suitable for VR geovisualizations. The major challenge is to achieve the application of the cartographic principles and scale issues of 2D and 3D maps and geospatial data when imported into a VR environment.

This paper studies the applicability of UAS data acquisition, flight characteristics, and the quality of the derived maps and 3D models of geoheritage areas in VR geovisualization. This research aims to develop an overarching methodology that utilizes cartographic scale for the geoheritage monuments' multiscale 3D mapping. The objective is to devise and implement UAS flight characteristics (flying altitude, pattern, and camera angle) to generate high-resolution 2D maps and 3D models sufficient for VR geovisualization across the three cartographic scales. The proposed UAS flight characteristics can be applied in various timeframes. This highlights the advantages and practicability of UAS in monitoring dynamic activities such as the formation, geoconservation, and promotion of sites with high geological significance.

The main contribution of this work is the utilization of cartographic scale principles for the creation of efficient visualizations in VR. Cartographic principles are applied, and the spatial resolution of images is correlated with the detail and precision of the 3D models produced from data collected via UASs. The flight altitude, path, and camera angle of the UAS affect both the geometry and the precision of the 3D models as well as the quality of the texture. More specifically, a geometrically complex object needs lateral shots to be captured with detail in 3D, while a flat area can be captured by only vertical shots. This, combined with the mapping scale and the eye's visual acuity, determines the resolution of the 3D model to efficiently geovisualize the area/object studied. The above parameters are important for visualizations in VR, as the user of the application has direct contact with the 3D model and can observe it circumferentially. This study focuses on the integration of UAS technology and VR techniques for an efficient multitemporal 3D geovisualization of geological monuments on various cartographic scales.

## 2. Materials and Methods

### 2.1. Study Area

The paper's study area is part of the wider area of the Petrified Forest of Lesvos in the northeastern Aegean Sea in Greece. It consists of a rare, petrified forest ecosystem with concentrations of petrified trees covered by volcanic material and fossilized in place millions of years ago. Petrified tree trunks, branches, roots, and tree leaves are revealed under layers of volcanic ash [70,71].

By Presidential Decree (PD 443/1985), the Petrified Forest was declared a natural monument. Every small or large part of the fossilized trunks in the area is protected by law. The Petrified Forest area was a founding member of the European Geoparks Network in 2000 and joined the UNESCO World Geoparks Network in 2004 [72,73].

The new road opening between Kalloni and Sigri brought new fossil sites to light. Several standing and fallen tree trunks were discovered alongside the road, especially in the western part. For this reason, the sites with a high concentration of fossils had to be optimally configured to preserve and protect the findings. Parks for visitors have been established in areas with a high concentration of petrified trees. This location is a representative example of the fossilite ferrous sites, containing petrified tree trunks that appear at different altitudes and an impressive root system (Figure 1). The area is located 3 km before reaching the settlement of Sigri, and its orientation is west.

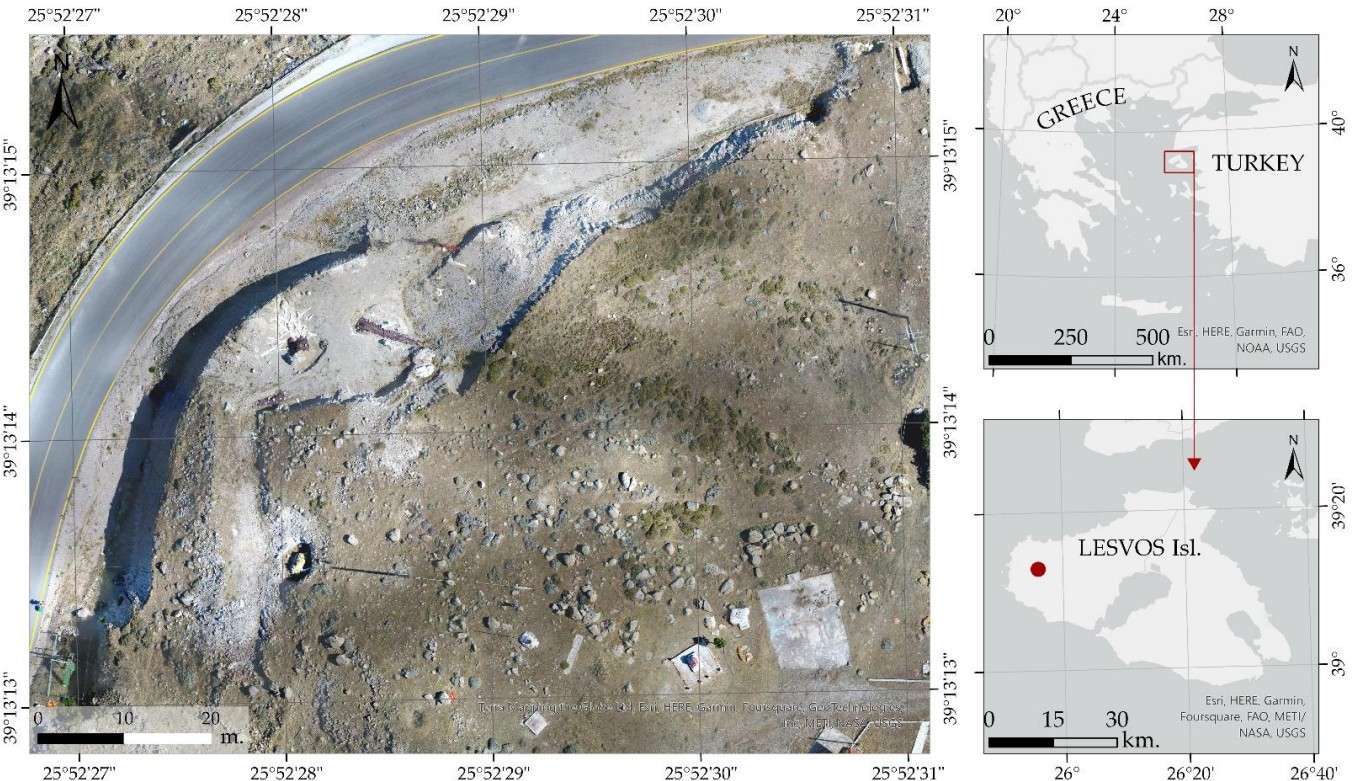

**Figure 1.** Location of the fossilite ferrous site, Lesvos Petrified Forest, Lesvos, Greece (Sources: Esri, HERE, Garmin, FAO, METI/NASA, USGS).

The Museum of Natural History of the Petrified Forest of Lesvos carried out procedures for protecting, conserving, and promoting fossils, making the site accessible to the public. The above management processes were completed at various time intervals, and the alterations in the area were obvious. Thus, the methodology presented in this work was developed to spatiotemporally monitor the alterations and the development of work on the fossilite ferrous site.

### 2.2. Methodology

The following methodology was developed to integrate UAS technology with VR techniques for the most efficient multitemporal 3D geovisualization of geological heritage on different cartographic scales (Figure 2). A scale investigation of the geographical area of the site is first conducted. More specifically, the three geographical scales are: (i) the fossil site (1:120), (ii) the fossil root system (1:20), and (iii) individual fossils (≥1:10). From the determination of the geographical scale, three corresponding cartographic scales emerged: (a) 1: 120, (b) 1:20, and (c) 1:10. The definition of 3 different cartographic scales leads to the

suitable resolution of the cartographic results. The ground sampling distance (GSD) of the very high-resolution images (VHRI) and the camera features of each UAS determined the flight altitude for each cartographic scale. The flight pattern and the camera orientation were then configured for each case. Along with the image acquisition, the ground control points (GCPs) were collected, and were used for georeferencing.

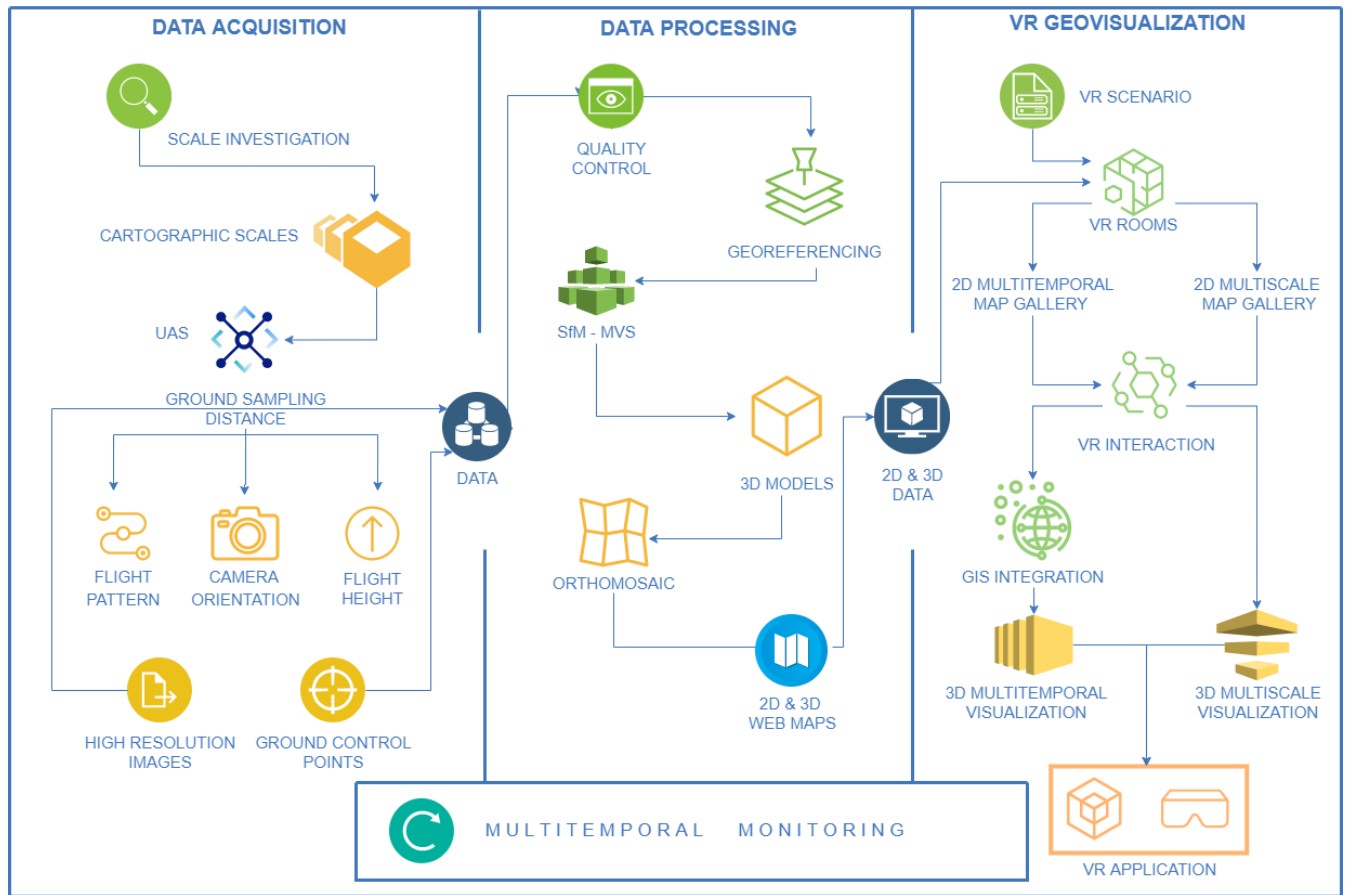

**Figure 2.** Flowchart of the main stages in the methodology.

The raw data collected were then inspected for their suitability and reliability in a post-process step. VHRIs were georeferenced with GCPs, and then the structure from motion and multiview Stereo algorithms were applied to create the 3D dense point cloud and the 3D model via Metashape by Agisoft [74]. Then, the digital surface model (DSM) and the orthomosaic of the study area were created. The data were acquired on eight different dates, simultaneously with the maintenance and promotion work in the fossil site area. The image-based 3D modeling processing results were imported into ArcGIS Pro software [75], where 2D and 3D maps were created. The 2D and 3D maps were then published online via ArcGIS Enterprise [76].

The next stage in the methodology examines the visualization of 2D and 3D multitemporal and multiscale results in a virtual reality environment. Initially, the VR application scenario was structured. The scenario comprises two virtual gallery rooms, one for the spatiotemporal geovisualization of the results and a second for the multiscale geovisualization. Specific interactive features between the user and the virtual space were programmed in both rooms, along with interactive features for each room. Some interactive features were combined with the geographical information systems (GIS) through a specific software development kit (SDK) that allows the transfer of real coordinates in virtual space. The application's functionality was tried and tested using VR equipment and was finally launched in an executable game format (.exe) compatible with the Windows operating system.

### 2.3. UASs and GNSS

Data were acquired from the area using four different UASs, with different recording sensor and lens features (Table 1). More specifically, the first UAS used was the Phantom 4 Pro by DJI. The DJI Phantom 4 Pro is a mid-range quadcopter aircraft (1375 g in weight) with a three-axis stabilized camera. The Phantom 4 Pro camera has a 1-inch, 20-megapixel CMOS sensor FC6310 and a manually adjustable aperture from f2.8 to f11. It also has a focus range from 1 m to infinite. The Phantom 4 Pro uses a camera lens optimized for aerial imaging, with a 24 mm equivalent focal length and 9 mm real focal length. The actual sensor size is 13.2 × 8.8, and the images are 3:2. The aspect ratio is 5472 × 3648 pixels. The second UAS was the Mavic 2 Pro, equipped with a Hasselblad camera. This camera has a 20-megapixel resolution and a 1" CMOS sensor. The sensor's dimensions are 13.2 × 8.8 mm, and the resolution of the images in the 3:2 ratio is 5472 × 3648 pixels. The Hasselblad apertures are f/2.8–f/11 and the lens is fixed with a true focal length of 11 mm. The body of this UAS weighs 907 g. In addition, it has a collision sensor and four propellers. The third UAS used for data collection was the Inspire 2, a quadcopter weighing 3400 g and belonging to the middle class of UASs. It has a gimbal with a three-axis stabilizer where different recording sensors can be placed. The recording sensor used for this task was the Zenmuse X5S camera. This camera has a CMOS sensor 4/3 and a resolution of 20.8 mpxl. The actual size of the sensor is 13 × 17.3, and the resolution of the images captured is 5280 × 3956 pixels. Different lenses can be mounted on the X5S. For the purposes of this recording, two lenses were used: (a) a DJI MFT with a focal length of 15 mm and an aperture of 1.7, and (b) an Olympus M.Zuiko with a focal length of 25 mm and an aperture of 1.8. The fourth UAS used for data acquisition was the Matrice 300, equipped with a Zenmuse P1 camera. The camera specifications are: (i) 48 mpxl resolution, (ii) f/2.8–f/16 aperture, (iii) full-frame recording sensor with 35.9 × 24 mm actual size, and (iv) 8192 × 5460 pixels resolution. The Zenmuse P1 supports various lenses with different focal lengths. In this study, a 35 mm lens was used.

**Table 1.** Characteristics of UASs and cameras used in image acquisition.

| UASs/Cameras | Megapixel | Aperture | | fr (mm) Equivalent | fr (mm) Actual | | iw (Pixel) | sw (mm) |
|---|---|---|---|---|---|---|---|---|
| Phantom 4 Pro | 20 | f/2.8–f/11 | | 24 | 8.8 | | 5472 × 3648 | 13.2 × 8.8 |
| Mavic Pro Hasselblad | 20 | f/2.8–f/11 | | 28 | 11 | | 5472 × 3648 | 13.2 × 8.8 |
| Inspire (Zenmuse X5S) | 20.8 | | | | | | 5280 × 3956 | 13 × 17.3 |
| DJI MFT　　　Olympus | | f/1.7 | f/1.8 | | 15 | 25 | | |
| Matrice 300 (Zenmuse P1) | 48 | f/2.8–f/16 | | | 35 | | 8192 × 5460 | 35.9 × 24 |

The Hiper SR receiver was used for the Global Navigation Satellite System (GNSS) measurements. This equipment consists of two parts, (i) the base and (ii) the rover, which communicate with each other. The Hiper SR receiver has 226 Channels for Universal Tracking and receives the signal from GPS and GLONASS. It can operate in 4 different modes: (a) Static/Fast Static, (b) Precision Static, (c) Real Time Kinematic (RTK), and (d) DGPS. For the present work, the RTK mode was chosen, the accuracy of which reaches up to 10 mm horizontally and up to 15 mm vertically.

### 2.4. Virtual Reality (VR)

Virtual reality techniques and equipment were used for the 3D geovisualization of the results obtained after processing the data collected with UAS. The VR headset was the Valve Index model. This package includes the VR headset, two controllers, and two base stations. The Valve Index Headset display includes stereo RGB screens, allowing immersion in the virtual world. The Headset has two LCD 1440 × 1600 screens, full RGB per pixel, and a super low total backlight stay (0.330 ms at 144 Hz). The response rate of the frames projected on the cameras is 80/90/120/144 Hz. They have an ergonomic design as they allow the adjustment of: (i) the interstitial distance of the convex lenses with

58–70 mm range, (ii) the distance of the lenses from the eyes (front–back), (iii) the size of the perimeter of the head, and (iv) the position of the headphones, with a natural fit. The Headset headphones have a balanced operation of 37.5 mm with a frequency response of 40 Hz–24 KHz.

The VR glasses work in combination with the two base stations and the two controllers. Valve Index controllers are compatible with any headset that supports SteamVR [77] detection. Each controller corresponds to either the left or the right hand and detects hand and finger position, movement, and pressure through 87 sensors to determine the user's intention. Valve Index controllers allow reaching out and grasping an object directly instead of relying on buttons such as triggers. In addition, they have a wrist strap that allows the opening of the palm so that objects can be thrown. The strap is easy to secure and fits a variety of hand sizes. The main body of each controller is the input of the handle, with built-in power sensors set to detect a wide range of forces from a gentle touch to a strong grip. This improves physical actions such as gripping and throwing objects and introduces new interactions such as tightening and crushing. The ten buttons on each controller can be programmed or deactivated depending on the needs of the game.

The package includes two base stations. Valve Index Base Stations are equipped with fixed lasers that scan 100 times per second to detect photon sensors on the headset and controllers. Range 2 base stations cover an area of $7 \times 7$ m, and their field of vision is $160° \times 115°$. These sensors are also compatible with other VR headsets such as the HTC VIVE. This equipment operates via Steam VR. SteamVR is a tool for experiencing virtual reality content that is compatible with Valve Index. This tool allows you to delimit the area where VR equipment is used and is fully compatible with game engine software such as Unity and Unreal engine.

## 3. Data Acquisition and Processing

### 3.1. Flight Planning

The first stage of data acquisition was scale investigation. A total of three geographical scales emerged, depending on the observation needs of the fossil site's management board. The largest geographical scale includes the entire fossil-bearing site. The site's configuration was carried out from 2018 to 2021, resulting in the need to monitor the work's progress and capture the area at different dates. The second geographical scale that emerged was at the root-system level, and the third scale was at the fossil level, where conservation procedures of the root and individual fossils were carried out. The two smaller geographical scales were not captured over time as the findings are covered with special protection material during fieldwork. The geographical extent of each case led to three different cartographic scales: (i) $\geq$1:10, (ii) 1:20, and (iii) 1:120. This defined the required GSD of the images acquired for the respective cartographic scales: (i) $\leq$0.1 cm, (ii) $\leq$0.20 cm, and (iii) $\leq$1.3 cm.

The GSD of the images and the camera's features determine the flight altitude of a UAS. In this case, three different aircraft and four different recording sensors were utilized, and the flight altitude was calculated separately for each camera. Table 2 lists the flight dates, the UASs and the recording sensor that collected the data, the cartographic scale, the GSD, and the corresponding flight altitude.

Data collection was performed on different dates between July 2018 and May 2022. In 2019, no flights were operated due to the absence of work for the site's configuration. In 2020, all maintenance and promotion processes ceased due to the COVID-19 pandemic. The first flight was executed on 11 July 2018, with a Phantom 4 Pro. This flight covered a large geographical area as its purpose was to define clear boundaries for the site (Figure 3a). The flight was executed at a 30 m flight altitude, and the GSD of the images was 0.8 cm. The GSD was less than 1.3 cm, so the data was suitable for a 1:120 cartographic scale. The overlap of the VHRI was 80% front and 70% side. The flight lasted 20 min in total, and 600 images were collected. The second flight took place on 13 March 2021, with a Phantom 4 Pro for a 1:120 cartographic scale (Figure 3b). Up to this date, the site limits had been set, so the mapping area was limited to the fossil-site level. The flight was accomplished

at 40 m, the GSD of the images was 1.07 cm, and the overlap of the images was 80% front and 60% side. The flight lasted 8 min in total. The third flight was performed on 8 May 2021, with the Inspire 2 and the Zenmuse X5S camera with the Olympus 25 mm lens, which mapped the area at a scale of 1:120. The flight altitude was 50 m, and the spatial resolution of the images was 0.5 cm. The overlap of the images was 80% front and 70% side, and 130 images were collected. The total duration of this flight was 7 min. On the same date (8 May 2021), the flight captured the area at the fossilized root-system level, i.e., at a scale of 1:20. The aircraft followed a pattern perimetric to the point of interest for 3D mapping at a 1:20 scale (Figure 3c).

**Table 2.** Recording dates, UAS type, cartographic scale, corresponding GSD, and flight altitude.

| Date-Month-Year | UAS | Recording Sensor | Scale | GSD | Height of Flight |
|---|---|---|---|---|---|
| 11/07/2018 | Phantom 4 Pro | DJI 20 MP (9 mm) | 1:120 | 0.8 cm (<1.3 cm) | 30 m |
| 13/03/2021 | Phantom 4 Pro | DJI 20 MP (9 mm) | 1:120 | 1.07 cm (<1.3 cm) | 40 m |
| 08/05/2021 | Inspire 2 | Zenmuse X5 S (25 mm) | 1:120 | 0.5 cm (<1.3 cm) | 50 m |
| 08/05/2021 | Inspire 2 | Zenmuse X5 S (25 mm) | 1:20 | 0.16 cm (<0.20 cm) | 15 m |
| 06/07/2021 | Inspire 2 | Zenmuse X5 S (15 mm) | 1:120 | 0.9 cm (<1.3 cm) | 40 m |
| 21/07/2021 | Mavic 2 Pro | Hasselblad (11 mm) | 1:120 | 0.16 cm (<1.3 cm) | 40 m |
| 11/09/2021 | Mavic 2 Pro | Hasselblad (11 mm) | 1:120 | 0.16 cm (<1.3 cm) | 40 m |
| 11/09/2021 | Inspire 2 | Zenmuse X5 S (25 mm) | ≥1:10 | 0.1 cm (≤0.1 cm) | 5 m |
| 13/10/2021 | Inspire 2 | Zenmuse X5 S (15 mm) | 1:120 | 0.9 cm (<1.3 cm) | 40 m |
| 15/05/2022 | Matrice 300 | Zenmuse P1 (35 mm) | 1:120 | 0.5 cm (<1.3 cm) | 50 m |

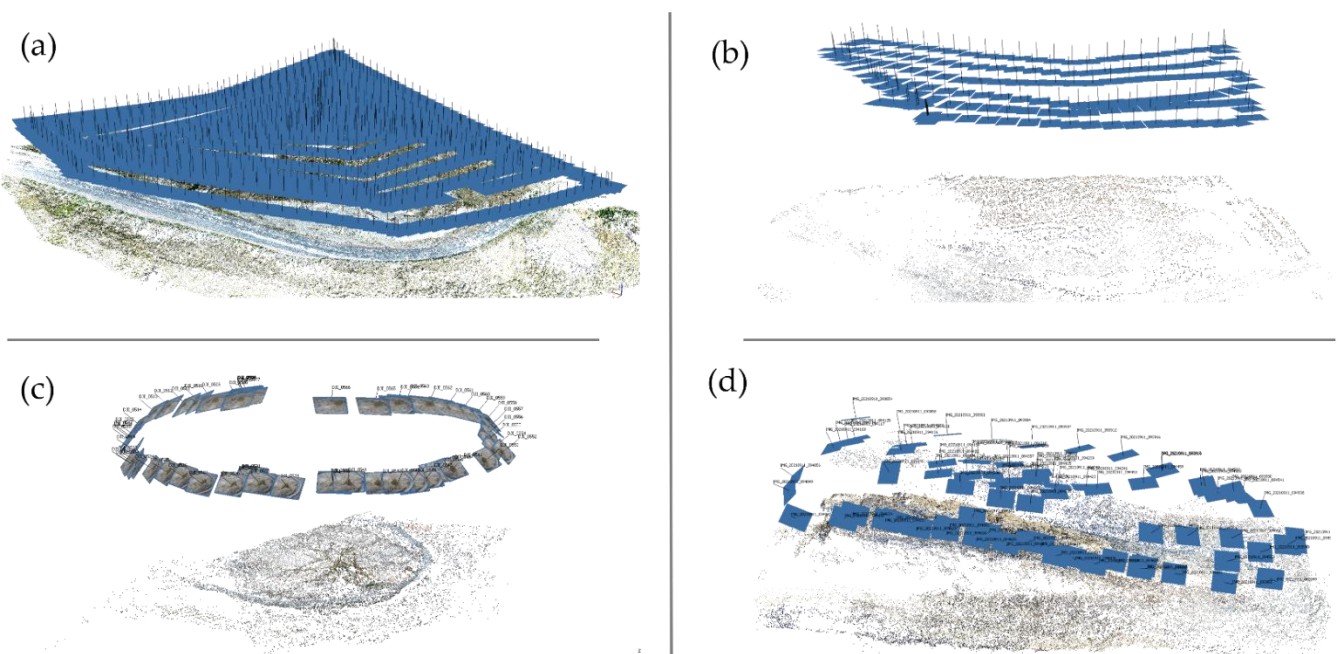

**Figure 3.** Flight plans for: (**a**) wider study area, (**b**) fossil-bearing site, (**c**) root system, and (**d**) single fossils.

On 6 July 2021, the Inspire 2 and the DJI MFT camera (15 mm) acquired data in a grid form and at a flight altitude of 40 m. The images had a GSD of 0.9 cm (<1.3 cm) and an 80% front and 60% side overlap. The next flight took place on 21 July 2021, with the Mavic 2 Pro, where the flight altitude was 40 m and the GSD 0.16 cm (<1.3 cm). An identical

flight with the same equipment was performed on 11 September 2021 and acquired data for mapping at a 1:120 cartographic scale. All flights were designed considering the area's high relief, with the aircraft flying at a constant distance above the ground. In addition, the camera was set at a −90° angle, i.e., vertical to the ground. On September 11, a very low flight was conducted at 5 m with the Inspire 2 and the Zenmuse X5S (25 mm) camera. This flight aimed to produce an orthophoto map and a high-resolution 3D photorealistic model at a cartographic scale of 1:10. The pattern followed was perimetric to the site with the camera in an oblique position (Figure 3d). Additional images were acquired perpendicular to the fossil at the same flight altitude. Work on the fossil-bearing site was also recorded on 13 October 2021. The flight was conducted with the Inspire 2 (DJI MFT, 15 mm) at 40 m flight altitude. The 160 images acquired had an 80% front and 70% side overlap. Their GSD was 0.9 cm (<1.3 cm). All the flights mentioned above were performed through the Litchi Mission Hub software [78].

The most recent recording date of the site was 15 May 2022. The Matrice 300 and the Zenmuse P1 (35 mm) camera were used. The flight of the Matrice 300 had an altitude of 50 m, and the images had a GSD of 0.5 cm (<1.3 cm), suitable for mapping the area on a 1:120 cartographic scale. The flight was designed with the DJI Pilot application [79]. The front overlap of the images was 80% and the side overlap was 60%. A total of 170 photos were collected and the flight's duration was 13 min. The study area presents several difficulties in data acquisition due to the intense elevation differences in the territory occupied by the fossil-bearing site. In addition, its northwest orientation creates intense shadows on the slopes and the steep sides inside the trenches in the morning hours. Another difficulty encountered during data acquisition was the dust emanating in the atmosphere due to the ongoing work at the site.

### 3.2. Georeferencing

VHRIs georeferencing was performed using ground control points (GCPs). The points were measured with the Hiper SR receiver and the RTK method [80]. The coordinates system used to measure the GCPs and georeference the VHRIs was the Greek Grid 2100. In more detail, 20 GCPs were measured in the surrounding area of the fossil site. The position points were collected on 11 July 2018. Figure 4 shows the study area calculations. The positions used as control points are displayed in red, and the checkpoints in yellow. In addition, checkpoints are concentrated on the southeast side of the area because defining the position's boundaries has been the main target since the first date of the position recording. Consequently, points that demarcated the wider area were acquired. Then, the monitoring area was limited to the north and west sections where fossils appear. Therefore, the VHRIs were georeferenced with the four red GCPs, located around the perimeter of the fossil site and at different altitudes. The same control points were utilized on all recording dates as they were located at established positions in the area. The internal measurements of the position were the checkpoints. However, these calculations could not be utilized on all dates as the maintenance and promotion work in the area was intense. More specifically, protective walls, paths, and individual positions were created in the area, resulting in the disappearance/loss of several checkpoints.

Based on the 1:120 cartographic scale, in which the entire fossilite ferrous site was mapped, the acceptable error of horizontal and elevation accuracy is 4 cm [80,81]. Table 3 shows the GCPs errors by x, y, z axis and the total RMS for each recording date for the 1:120 cartographic scale. More specifically, the total RMS of the data georeferencing was: (i) 1.74 cm on 11 July 2018, (ii) 3.12 cm on 13 March 2021, (iii) 2.73 cm on 8 May 2021, (iv) 3.54 cm on 6 July 2021, (v) 3.53 cm on 21 July 21, 2021, (vi) 3.38 cm on 11 September 2021, and (vi) 4.06 cm on 13 October 2021.

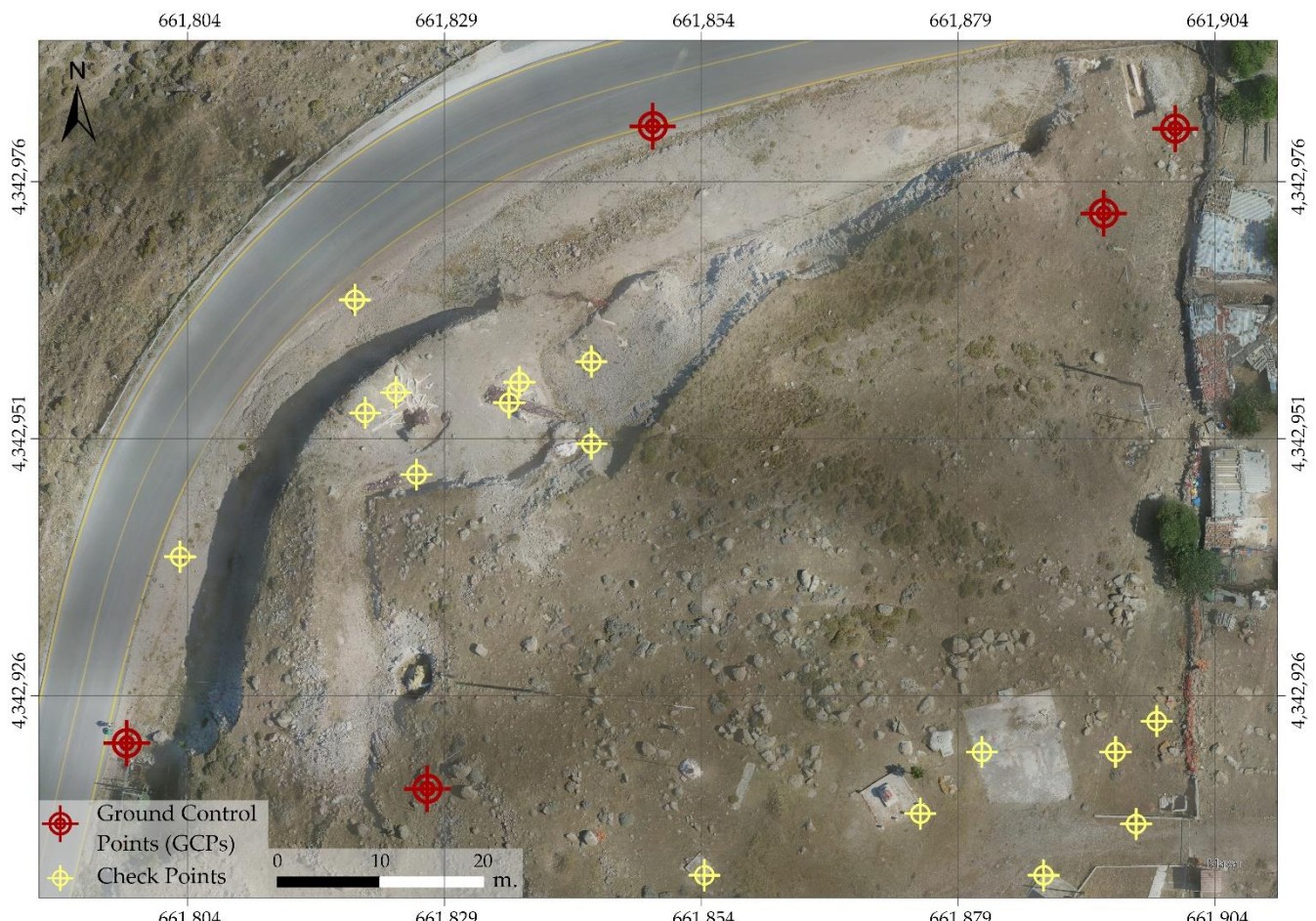

**Figure 4.** Map with the measurements' positions of the ground control points and the checkpoints.

**Table 3.** Georeference error table on x, y, z, and total RMS for all datasets collected on eight recording dates.

| Date-Month-Year | X Error (cm) | Y Error (cm) | Z Error (cm) | Total RMS (cm) |
|:---:|:---:|:---:|:---:|:---:|
| 11/07/2018 | 0.40 | 0.93 | 1.41 | 1.74 |
| 13/03/2021 | 2.48 | 1.82 | 0.52 | 3.12 |
| 08/05/2021 | 2.42 | 1.24 | 0.24 | 2.73 |
| 06/07/2021 | 2.02 | 2.78 | 0.84 | 3.54 |
| 21/07/2021 | 2.66 | 2.05 | 1.06 | 3.53 |
| 11/09/2021 | 2.44 | 2.11 | 1.03 | 3.38 |
| 13/10/2021 | 2.60 | 3.01 | 0.80 | 4.06 |
| 15/05/2022 | 2.59 | 2.97 | 1.1 | 4.01 |

At the last recording date on 15 May 2022, the equipment consisted of the Matrice 300, a UAS with a built-in RTK receiver. Thus, in combination with a D-RTK DJI, it is possible to acquire primary data with very high accuracy. The D-RTK was placed at a point with known coordinates and created a network between the aircraft and the reference (D-RTK base), which helped achieve an error of 4.01 cm. The data coordinates were then transferred from the World Geodetic System 1984 (WGS 84) to the Greek Grid 2100 projection system. The horizontal and altitude accuracy of the data acquired with the Matrice 300 were checked by GSPs placed in the area on previous dates.

*3.3. Image-Based 3D Modeling Process &Results*

According to the methodology, the next stage was data processing to create a 3D point cloud, 3D models, and very high-resolution orthomosaics. The data were processed with photogrammetric and image-based 3D modeling methods to produce the cartographic results. The steps followed for image-based 3D modeling processing were the same for all datasets at each recording date.

Initially, quality control of VHRIs was performed visually, first, by an expert photo interpreter, and then, using the Image Quality Index (IQI) algorithm [82]. Images that were blurred, shaken, overexposed, or included parts of the horizon were excluded through visual controls. Then the VHRIs that showed values outside the limits of 0.5–1 in the IQI index were rejected through the next steps in processing. The VHRIs suitable for photogrammetric processing were introduced in the Agisoft Metashape [74] software, where the alignment of the images was applied. The alignment process includes the implementation of structure from motion (SfM) [83,84], which consists of two algorithms, the scale invariant feature transform (SIFT) [85,86] and the random sample consensus (RANSAC) [87,88]. The application of the above in VHRIs results in a sparse point cloud. A denser point cloud is then generated using the multiview stereo (MVS) algorithm [83,89]. The resulting 3D dense point cloud is the basis for creating a 3D mesh. Specifically, through spatial interpolation, the points are connected in a triangulation irregular network (TIN) which consists of a single 3D mesh. The 3D mesh is enriched with a photorealistic texture, and a textured 3D model emerges. This is followed first by the creation of the Digital Surface Model (DSM), which describes the altitude of the area, and finally by the orthorectification of the images' pixels to create the orthomosaic. This processing was applied to the data acquired on the eight recording dates for each of the three different scales. The image-based 3D modeling processing results utilized to develop the VR application were 3D models and orthophotomaps, as presented in Table 4.

More specifically, the 3D model and the orthophotomap produced for 11 July 2018 included two standing petrified tree trunks. The part of the area limited by the fence was parallel to the road, and the location was not yet accessible to the public. On the next recording date 13 March 2021, increased activity due to work resulted in changes. There were transported lying petrified tree trunks, the fossils had been covered with a special protective material, and the first demarcation fences for each site were installed. On 8 May 2021, the fossil ferrous site margins were designated with the wall raised around it. At the beginning of July in the same year, works in the fossilized root system as well as in various standing tree trunks of the study area were recorded. By the end of 21 July 2021, the site had assumed a distinctive form as the viewing level of the petrified trunks was comprehensible. The first paths had been built, numbered signs had been placed on each find, and trunks had been transported in a suitable display position. In addition, the fence had been considerably extended towards the side of the road. On 11 September2021, the construction of a staircase in the northwestern part of the site allowed access to its highest level. In addition, the equipment and tools had been removed from the area. The vegetation had been removed from the paths, and the site and the construction of the stone walls had been completed. On the last date of mapping of the fossil site for 13 October 2021, wooden fences and stairs appeared at the perimeter of the findings. On 13 October 2021, all works for promoting and maintaining the fossil site were completed, and the site was fully configured for the public to visit. A flight was performed in the spring of 2022 to capture the site's state a few months after the completion of the works.

**Table 4.** Indicative results of image-based 3D modeling processing: (A) 3D models, (B) orthophoto maps, and (C) recording dates.

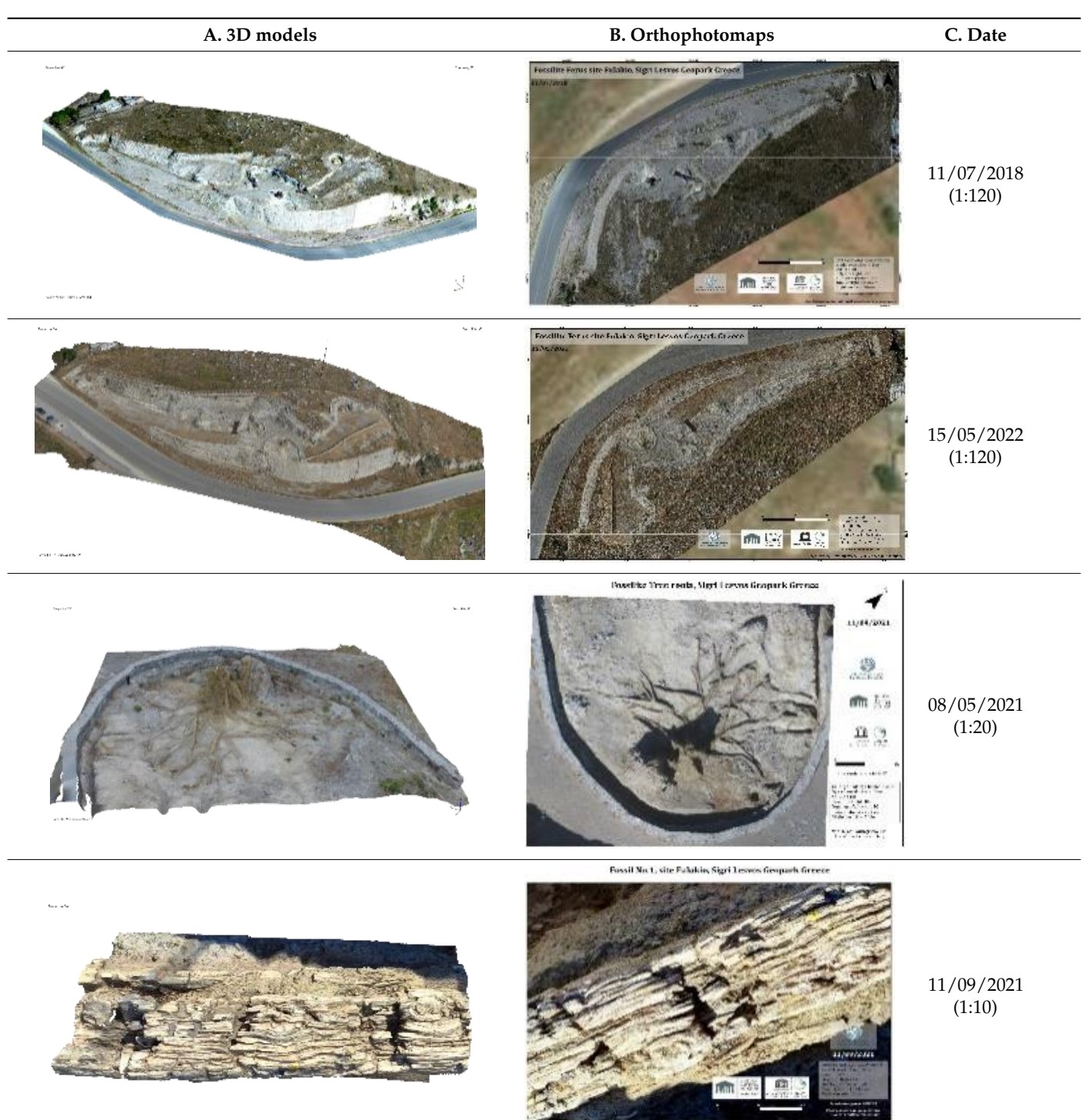

| A. 3D models | B. Orthophotomaps | C. Date |
|:---:|:---:|:---:|
| | | 11/07/2018 (1:120) |
| | | 15/05/2022 (1:120) |
| | | 08/05/2021 (1:20) |
| | | 11/09/2021 (1:10) |

At this point, the site was accessible to the public, as the protective material placed during the winter months had been removed from the fossils. Low vegetation was present, as the images used in the orthophoto map were acquired during spring on 15 May 2022.

The two models and two orthophotomaps created for the larger cartographic scales were: (i) at the root-system level (1:20) and (ii) at the fossil level ($\geq$1:10). The 3D model created on a 1:20 cartographic scale depicts the condition of the fossil on 8 May 2021, immediately after the protective material's removal. This model allows the inspection of

the root system and works supportively in its maintenance process. In addition, it functions as a log file for monitoring over time. The orthophotomap shows the root's structure and branches as well as the parts with erosion risk. The 3D model of the individual fossil visualizes the find with very high resolution and in detail. Due to the high spatial resolution of the VHRIs, the texture and colors of the fossil are rendered with great photorealism, while parts of the bark and its grooves can be observed. The lying petrified trunk selected for mapping at a scale of 1:10 has the unique characteristic of being transferred to the fossil–ferrous site under study after being found at a nearby geographical location. Since it can be transported, its high-resolution mapping was important for its maintenance and conservation.

### 3.4. 3D Point Cloud and 3D Models Evaluation

Table 5 presents data and information regarding the 3D models created for the fossil site, root system, and fossil trunk. In particular, it is observed that the 3D point cloud and the number of the 3D model's surfaces created on 11 July 2018 capture the wider region, thus a larger geographical area. For this reason, the specific 3D model consists of more surfaces compared to the rest of the models at the same cartographic scale (1:120). The only 3D model showing an even higher number of points and surfaces is the one created from the images collected on 15 May 2022. This occurs because the 2022 flight was performed with the Matrice 300 and the Zenmuse P1 (35 mm) camera. The Zenmuse P1 camera has a resolution of 48 mpxl, while the rest of the cameras used in this study have resolutions up to 20.8 mpxl. The higher resolution of the camera affects the number of points in the 3D dense point cloud, and this in turn determines the number of mesh surfaces. According to the table, 3D point clouds and models created for the cartographic scales of 1:20 and 1:1 gather a high number of surfaces and points. Compared to point clouds and 3D models for the 1:120 cartographic scale, they have fewer points and surfaces, but proportional to the area depicted in each case, the density of points and the detail in the surfaces is significantly higher.

**Table 5.** Number of points of 3D point clouds and number of surfaces of 3D meshes for each recording date and the corresponding cartographic scale.

| Date | Scale | Point Cloud | Faces |
|---|---|---|---|
| 11/07/2018 | 1:120 | 31,438,368 | 4,978,312 |
| 13/03/2021 | 1:120 | 20,729,534 | 3,213,255 |
| 08/05/2021 | 1:120 | 23,517,312 | 4,115,665 |
| 08/05/2021 | 1:20 | 6,354,596 | 2,644,784 |
| 06/07/2021 | 1:120 | 14,422,086 | 2,011,447 |
| 21/07/2021 | 1:120 | 6,065,784 | 2,856,779 |
| 11/09/2021 | 1:120 | 14,572,981 | 3,335,834 |
| 11/09/2021 | 1:10 | 14,195,113 | 4,947,022 |
| 13/10/2021 | 1:120 | 23,302,524 | 2,525,060 |
| 15/05/2022 | 1:120 | 33,875,313 | 6,273,056 |

To evaluate the 3D point clouds and 3D models, a comparison was made between the point clouds produced from images collected by UASs to generate 3D geovisualizations at 1:120, 1:20, and ≥1:10 scales. In essence, the 3D geovisualizations created for the three cartographic scales were compared. Comparisons were made between the point clouds as they formed the basis for the subsequent creation of the 3D models. The greater the number of cloud points, the greater the number of triangulation irregular networks (TIN) that make up the mesh. The larger number of surfaces describes the geometry of the area mapped (root system and fossil) in more detail. Therefore, for 3D mapping at large cartographic

scales such as 1:20 and ≥1:10, a large number of points in the cloud is required to result in a high-resolution 3D mesh. The comparison of point clouds was performed in the Cloud compare software (CloudCompare 2.11, 2020) and the methods used were Cloud to Cloud (C2C) distance and surface density.

Initially, the densest point cloud created from data collected for the cartographic scale of 1:120 (15 May 2022) was selected. From this point cloud the parts of the petrified root system and trunk were isolated. Then a comparison was made between the clouds of (i) RS-1: root system (1:20)–RS-2: root system (1:120) and (ii) PTT-1: petrified trunk (1:1)–PTT-2: petrified tree trunk/fossil (1:120), using the C2C distance and surface density methods.

Figure 5 shows the results of the comparison between the point cloud of the root system created from the data from 8 May 2021 (Figure 5d), and the point cloud created by the images collected on 15 May 2022 (Figure 5e). Figure 5a presents the results of the C2C method. The densest point cloud, namely the RS-1 cloud, was defined as the reference cloud and the algorithm was configured to locate points between the clouds at a distance of 2 cm. This distance was chosen as the desired mapping scale is 1:20. Moreover, the biggest differences between RS-1 & RS-2 are observed in the standing part of the petrified tree trunk, while in the flat part of the root system the distances between the two clouds are smaller, due to the way in which the images for 3D mapping were collected. The images used to create the RS-1 were oblique, thus achieving a more detailed 3D imaging of the entire petrified root system, including the standing part. On the contrary, the images utilized for the creation of RS-2 were vertical to the ground, resulting in the consequence that the complex geometric features of the petrified root system were not fully captured. In Figure 5a, it is observed that in the areas depicted in red, the distance between the point clouds is 2 cm or more. It is therefore concluded that the compared cloud (RS-2) has fewer points in the red regions than the reference cloud RS-1. This observation is also verified by the results of the surface density applied separately to each point cloud (RS-1 & RS-2). Figure 5b presents the surface density of RS-1. The cell size with which the surface density was applied was $2 \times 2$ cm. The areas in bright orange indicate the areas with a higher concentration of points. The largest part of the point cloud of the root system shows a high concentration of points, from 54,000 to 72,000. In addition, there are parts of the RS-1 containing up to 142,178 points per 4 cm$^2$. Figure 5c depicts the surface density of the RS-2 per $2 \times 2$ cm. The cloud appears sparser, as the geometry of the root system is not clearly discernible. The larger part of the RS-2 shows a point density of 800 to 3200 dots per 4 cm$^2$.

Equivalent comparisons were made for the point clouds of the individual petrified tree trunk, PTT-1: petrified tree trunk/fossil (≥1:10)—PTT-2 petrified tree trunk/fossil (1:120). Figure 6a shows the results of the C2C distance algorithm, which was applied for distances under 2 cm. The densest point cloud, i.e., PTT-1, was set as the reference cloud. The red areas appearing on the lower part and the side parts of the trunk as well are interpreted as a low number of points in the PTT-2 cloud. This occurs as the images of 15 May 2022, used to create PTT-2, were captured vertical to the ground, excluding the side parts of the fossil. On the contrary, the images for the mapping of the trunk at a cartographic scale of 1:10 were both vertical and oblique, as well as being collected from a much lower flight altitude (5 m). This significantly affects the surface density. Characteristically, the density of PTT-1 points is displayed in Figure 6b, where the largest part of the fossil shows 2500 to 4500 points per 4 cm$^2$, whereas the PTT-2 surface density shows 2,000,000 to 3,200,000 points per 4 cm$^2$. Finally, the total number of points of PTT-1 (Figure 6d) is considerably larger (11,342,955) than that of PTT-2 (14,645). The number of points significantly affects the 3D models' solidness. In the 3D mapping of complex geometric features such as a petrified tree trunk, a high number of points is required to achieve its accurate 3D representation. The high resolution of the 3D model in this case is important for the fossils' conservation, protection, and promotion.

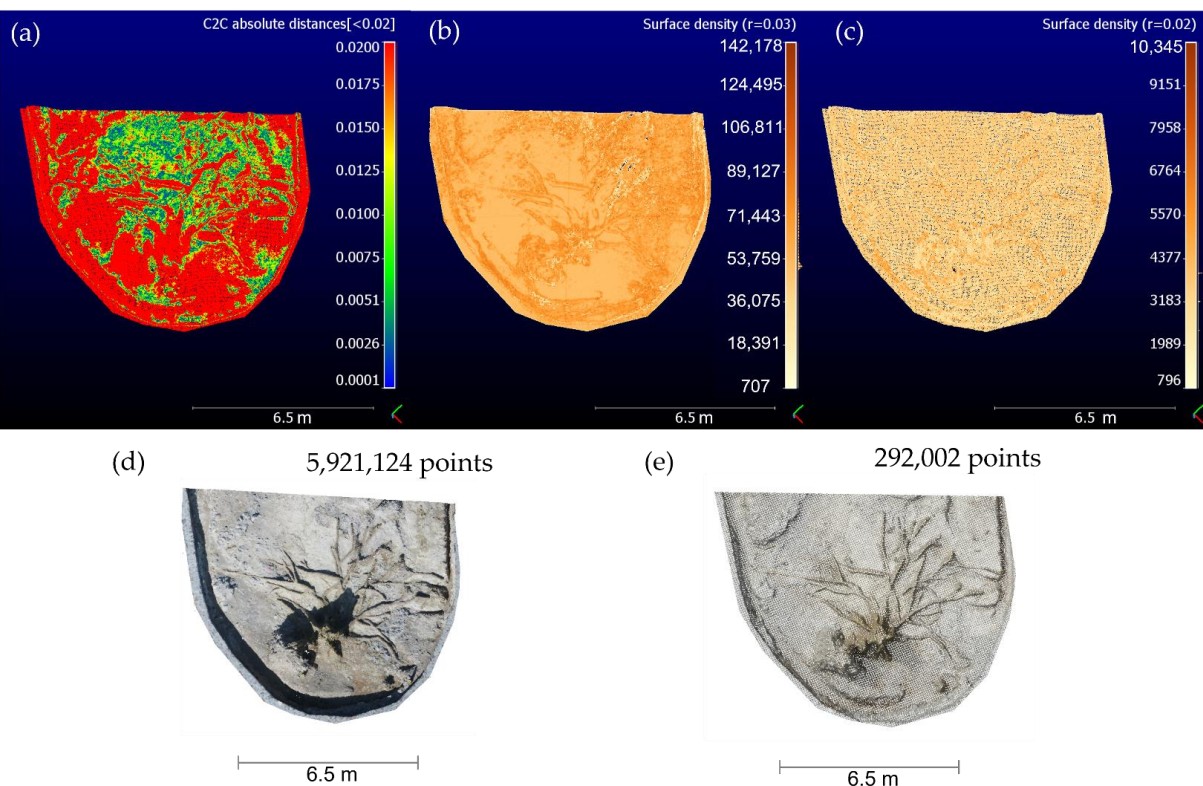

**Figure 5.** Comparison of point clouds of the petrified root system: (**a**) cloud to cloud distance, (**b**) surface density of RS-1, (**c**) surface density of RS-2, (**d**) RGB point cloud of RS-1 and (**e**) RGB point cloud of RS-2.

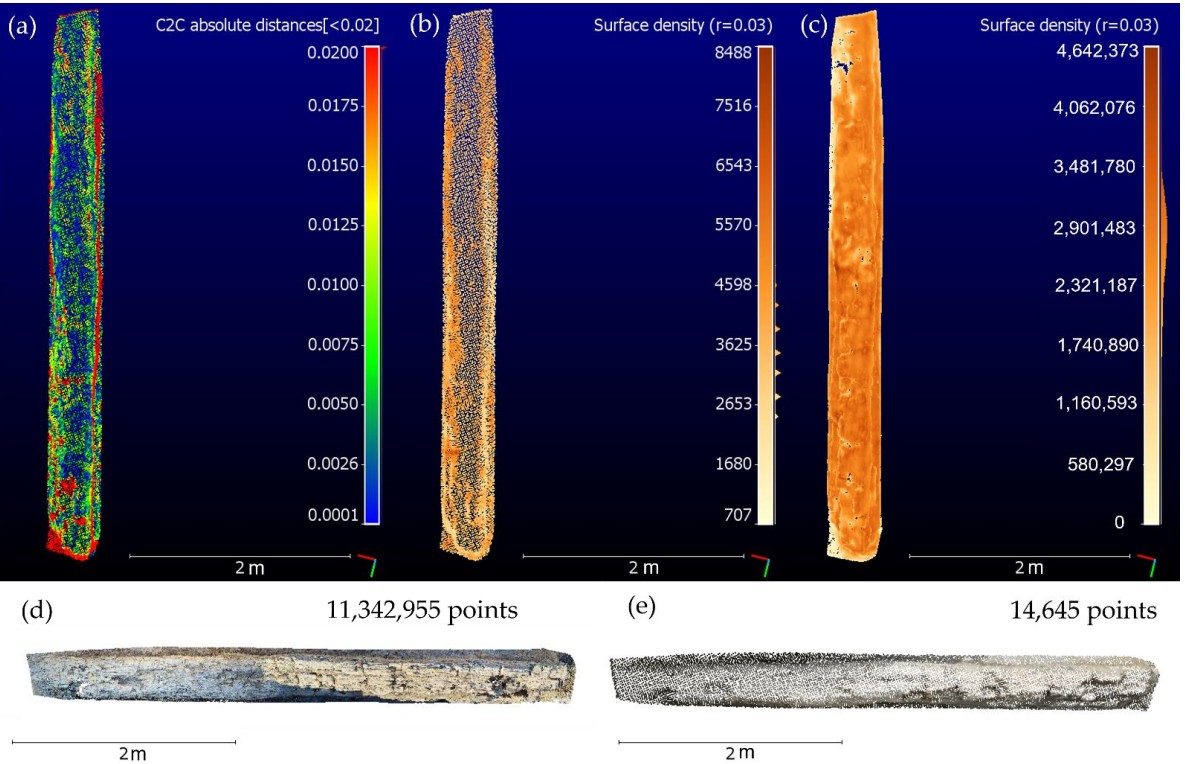

**Figure 6.** Point clouds of the fossilized trunk comparison: (**a**) cloud to cloud distance, (**b**) surface density of PPT-1, (**c**) surface density of PPT-2, (**d**) RGB point cloud of PPT-1, and (**e**) RGB point cloud of PPT-2.4. VR Geovisualization.

## 4. VR Geovisualization

### 4.1. VR Scenario

The design of VR applications is based on a scenario. The scenario is formulated to thoroughly cover the information that needs to be transferred through the virtual navigation [90,91]. The VR geovisualization of the spatiotemporal monitoring of the maintenance works and the promotion of the fossilite ferrous site studied in the present work was based on the scenario presented in Figure 7. The application developed supports a single user, but more observers can be added through suitable programming. This scenario had to meet the need for the geovisualization of two different but interrelated thematic units: (a) multitemporal and (b) multiscale. The main idea was the creation of two rooms, one for each thematic unit, one for the spatiotemporal monitoring of the site at a cartographic scale of 1:120, and another for the 3D geovisualization at various scales.

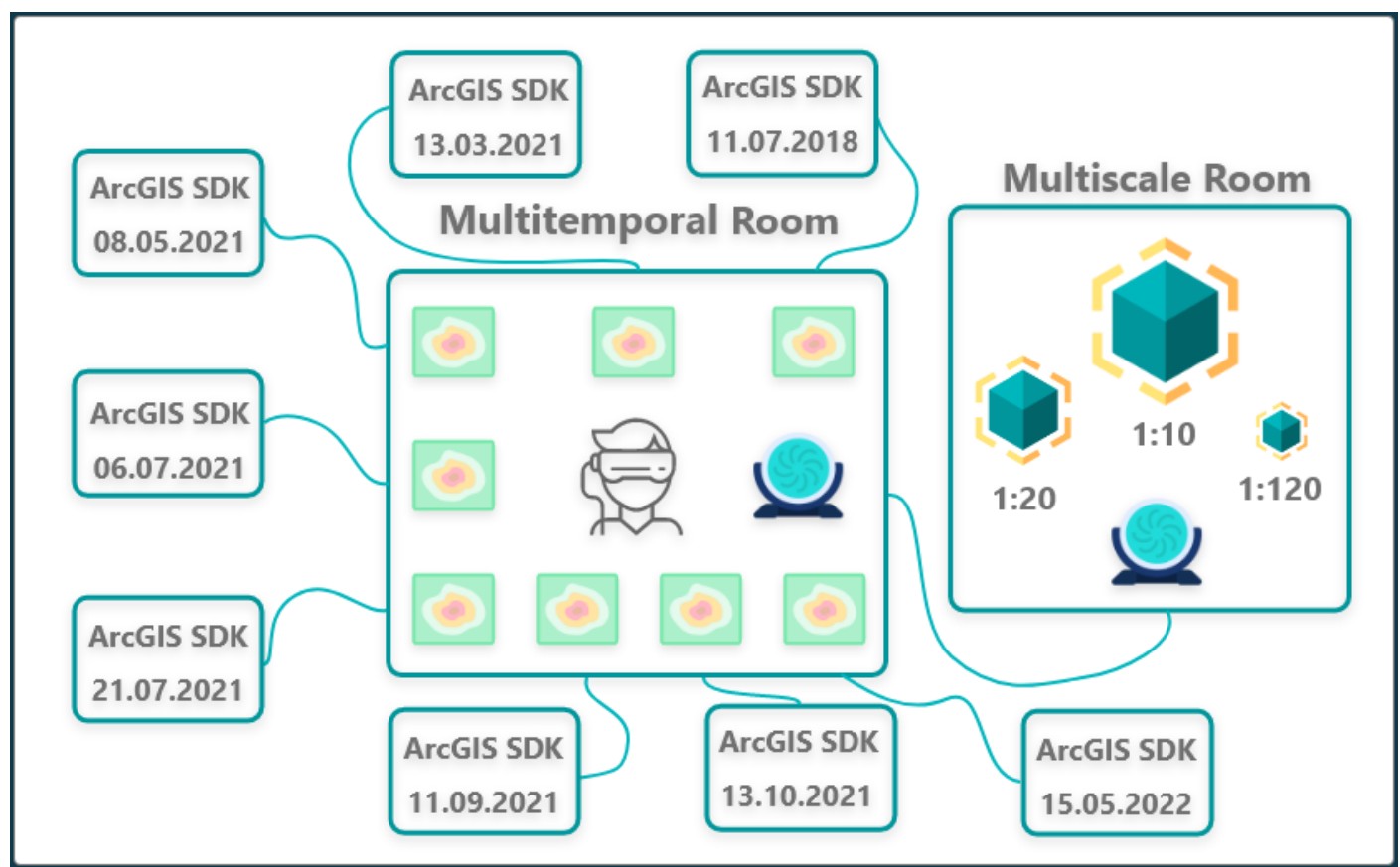

**Figure 7.** Structure of rooms/scenes in the VR application.

Firstly, a map gallery room was designed. Panels were placed at the room's perimeter showing the orthophoto maps created on a 1:120 scale for all recording dates. The maps were arranged in chronological order, beginning at the map created on the oldest recording date (11 July 2018) to the most recent (15 March 2022). The user can interact with the orthophoto maps and move to a different room through the panels. Each room concerns the corresponding date and the corresponding 3D imprint of the fossil site. The user observes the fossilite ferrous site from the corresponding flight altitude at which the UAS flew for the needs of a cartographic scale of 1:120. In the multitemporal room, a map contains all the geosites of Lesvos Island, emphasizing the fossil site located in the western part of the island. In the same room, a portal allows transfers from and to the room with multiscale geovisualizations. The second room was designed with the same pattern. The multiscale room was smaller, with three panels containing the maps corresponding to three different cartographic scales: (i) 1:120, (ii) 1:20, and (iii) ≥1:10. The maps were arranged from the

smallest to the largest cartographic scale. The user can interact with the maps by changing their format from 2D visualization of the area to its respective 3D visualization. A portal in this room allows the user to be transferred to the multitemporal room.

The virtual tour begins at the center of the multitemporal room, where most orthophotomaps of the study area are located, based on the time of data collection for their creation. The user has the ability to deliberately move in the space and observe any panel, although practically, they are guided through the chronological order of the maps. The first image is the map of Lesvos for the user's orientation regarding the geographical location of the fossil site. The timeline moves clockwise and ends with the most recent date. The portal at the end of the timeline contains an indication for moving to the multiscale room. In the second room, the user can move independently inside the space. However, the imaginary guide from the smallest to the largest cartographic scale makes the transition from the general to the individual comprehensible.

### 4.2. VR Processing and Programming

The 2D and 3D cartographic derivatives of the image-based 3D modeling process were used in creating the VR application. The 3D models and orthomosaics created in the Agisoft Metashape software [74] were exported in 3D object format (.odj) and raster format (.tiff), respectively. The orthomosaics were introduced into the ArcGIS Pro software (version 3.0.0) for cartographic process and composition. The process of cartographic composition resulted in 10 orthophotomaps which were then used to develop the virtual reality application. The 3D models were utilized in two ways to develop the application. The first was through ArcGIS Pro [75] and the second was as original 3D models (meshes). Regarding ArcGIS Pro [75], the 3D grid, materials, and jpg of each 3D model were introduced into the software and published on ArcGIS online. Through ArcGIS online, it is possible to share information on the web. Using a specially designed SDK, the 3D geospatial information on an ArcGIS server shared through ArcGIS online can be synchronized with a game engine. In the present work, the synchronization was executed with the Unity 3D Engine [92], specifically, version 2021.2.5f1. The SDK was used to create eight different rooms, each one corresponding to the eight different dates of recording at the fossil bearing site. As shown in Figure 8, the geographical location of the study area and the observer's position were stated with coordinates. In addition, the height from which the user observes the area under scale (1:120) was defined, as well as the camera's roll, pitch, and yaw. Then the relief was determined, and the Base Map was selected. Satellite mode was chosen as the base map, as the models are visualized in a photorealistic way. The geographical data, i.e., the 3D models that have retained their true coordinates, are imported into Unity 3D in layer form. The layer is literally URLs created through ArcGIS online and correspond to the 3D models of the area.

The virtual reality (VR) graphical interface was developed through Unity 3D Engines with the support of the Steam VR Platform for experiencing VR content on an Index Valve headset. Two base stations were integrated during Valve Index equipment installation. The base stations delimit the area in which the user can move autonomously. The area's dimensions were $3 \times 3$ m. Initially, two virtual scenes were designed to include the 2D and 3D maps and the 2D and 3D models that form the virtual room. The 3D orthophoto maps and models were positioned in the virtual room based on the script. Following the placement of all the objects in the virtual space, the stage lights also had to be adjusted. The scene's lighting was performed/determined with Auto Generate Lights for the homogeneous distribution of the lighting in the room. Simulation of the user in the virtual world is a key element in the design of the VR application; thus, an avatar that imitates the player's movements and actions is placed in the virtual space. The virtual room extended over an area of $10 \times 10$ m, where teleportation areas and teleportation points allowed the player to navigate the entirety of the area via the controllers.

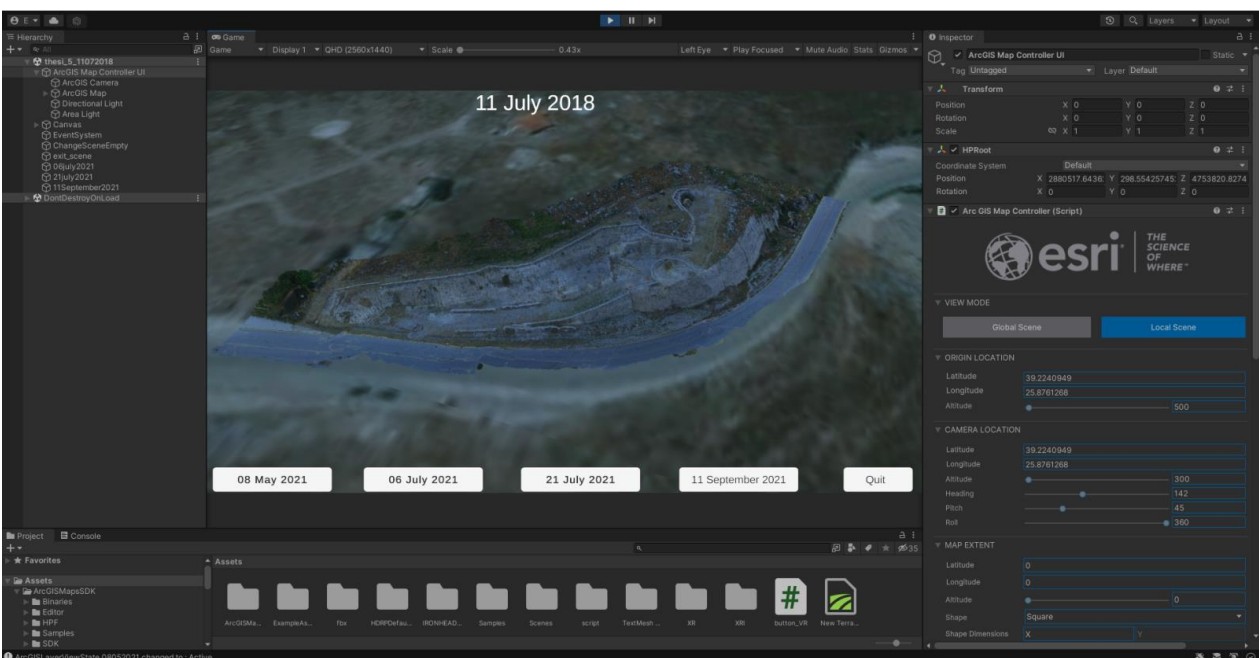

**Figure 8.** ArcGIS Maps SDK in Unity 3D Engine and the preview of fossilite ferrous site on 11 July 2018.

Teleport areas and Points were placed in the multitemporal and the multiscale scene. A perimetrical collider rendered a human form to the avatar's virtual body, so that the user could circumvent other objects in the scene. Additionally, maps triggered new actions when touched by the user. The first interaction programmed was alternation between the two scenes. This interaction was implemented with a script written in C# language, offering further properties to each panel and connecting it with the corresponding scene. This property was also utilized in a portal to allow transfer from the multitemporal to the multiscale room and vice versa. The multiscale room offers an additional action at the player's disposal. The action implemented a laser point for transition from the 2D orthophotomaps on the panels to the corresponding 3D orthophotomaps. This is achievable through the laser pointer on the controllers, where a C# script is activated by the trigger button.

## 5. Results—VR 3D Mapping

The result of this study is a VR application for the spatiotemporal monitoring of the maintenance and promotional works on a fossil site of the Lesvos Geopark at three different cartographic scales. The final application was launched in .exe format for user testing. Figure 7 shows the application in its entirety and the way it operates.

In more detail, Figure 7b shows the first room where the orthophotomaps' panels are displayed on a 1:120 cartographic scale for all recording dates. The player starts at the center of the multitemporal room viewing the Lesvos Island map, and is allowed to move voluntarily in a limited space of 3 × 3 m. The first virtual room is larger than 3 × 3 m, so a teleport area was placed at its center, along with individual teleport points. The teleport points were placed in front of the fossil site's orthophoto maps, as a way for the users to focus on the orthophoto maps without distractions from the surrounding space. In addition, the virtual player's body and head positions were adjusted to face the panel at a distance that allowed the entire map to cover their field of view. The maps were printed on A0 paper-size dimensions to maintain the appropriate spatial resolution (1.3 cm) for the 1:120 cartographic scale. Thus, the user can observe the changes throughout the different recording dates. Instructions on the panels offer information on the ways they can interact

with them. By touching the image of the map, the user moves to a different virtual space. This space is a 3D geographic area created by the ArcGIS Maps SDK.

A total of eight rooms were created using the ArcGIS Maps SDK. There are certain limitations concerning the eight rooms that correspond to the geovisualization of different timeframes. The ArcGIS Maps SDK operates with geographical coordinates in the WGS 84 reporting system, which leads to difficulties in navigation, as the user must be placed in a position with specific azimuths and at an absolute altitude. This difficulty was addressed by using the UAS coordinates and the ω, φ, κ angles of the camera, which were used as basepoints to place the observer in the 3D virtual geographic space. In rooms with different timeframes, the observer monitors the study area with no option to change the height to keep the observation scale constant at 1:120. To exit the rooms where the ArcGIS Maps SDK is used, the B button was programmed on both controllers to lead the user to the multitemporal room. A portal activated by touch allows the transition from the multitemporal to the multiscale room. The multiscale room is shown in Figure 9a. This room contains three panels, making it smaller than the multitemporal room. The three orthophotomap scales for the panels are: (i) 1:120, (ii) 1:20, and (iii) ≥1:10. The user can move voluntarily by walking in the room or by using the teleport points in front of the maps. Interaction with orthophotomaps in this room is executed with the use of pointers. A pointer is placed on the left controller, activated through the trigger on the lower part of the handle. Aiming at each panel and pressing the trigger, the orthophotomap is replaced by the corresponding 3D model (Figure 9c,d). The models selected for the three 3D mapping scales were: (i) the entirety of the fossil-bearing site (13 October 2021), (ii) the root system, and (iii) a single petrified tree trunk. Colliders were defined in each panel and model so that they will not be penetrated by mistake and to also enable switch interaction. Figure 9c and 9b shows the user's interaction with the 3D models. The user is able to approach the 3D models to explore and observe them from all sides and receive important data about the area and the findings, as the resolution of the 3D models and the textures is very high, accurate, and detailed. The addition of a portal similar to the one in the multitemporal room enables the transfer between the two rooms.

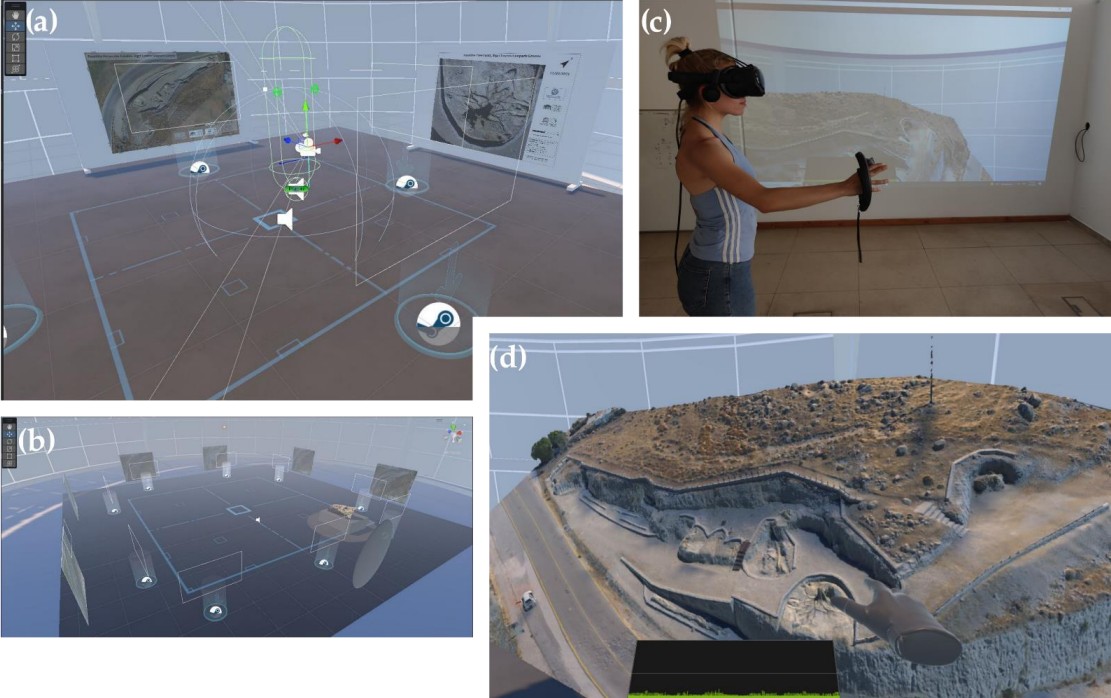

**Figure 9.** VR application: (**a**) multiscale room in Unity 3D Engine, (**b**) multitemporal room, panels, teleport area, and teleport points, (**c**) a user of the VR application, and (**d**) the corresponding image projected to the user in the VR headset via SteamVR.

## 6. Discussion

Every stage of the presented methodology raises important remarks. In the data acquisition stage, the topography of the study area affects the design of the flight plan. This observation applies to all UAS types utilized to acquire VHRIs. In addition, the high-relief areas significantly affect the primary data as the shadows are intense. To address this issue, the flights were performed at different times of day, depending on the season of the year. At this point it is worth mentioning that the use of different UASs is feasible for spatiotemporal monitoring at a constant mapping scale if the corresponding flight altitude is calculated for each recording sensor.

The results of the image-based 3D modeling process highlight the significance of the GSD of VHRIs for the orthophotomaps and the 3D models' reconstruction. More specifically, data acquired at a flight altitude of 40–60 m produce orthophotomaps and 3D models at a scale of up to 1:120. When the orthophotomaps and the 3D models of the fossil site are imported in a VR environment, close observation is inadequate for the accurate and precise depiction of the root system and the single located fossil. For this reason, it is crucial for the flight characteristics of the UAS to adapt to the mapping scale, and the geometry of the root system and the fossil. The root system was 3D mapped at a flight height of 15 m, providing an orthophotomap at a cartographic scale of 1:20. This scale allows the user to observe the root system in its entirety, and to distinguish root geometry without losing details. This provides an enhanced understanding of the 3D representation of a fossilized root system and aids the management board in observing the finding's pathogenesis to maintain it accordingly. For the high-resolution 3D mapping of the individual fossil and for its geovisualization in VR, VHRIs must be acquired at a very low flight altitude ($\leq$5 m). The 1:10 mapping scale in the virtual space thoroughly provides the fossil's geometry and texture. The geovisualization of the petrified tree trunk is thus both impressive and helpful, as every detail can be observed.

In the VR geovisualization stage for the integration of game engines, GIS and UAS are performed. This process requires high computational power and knowledgeability in the use of corresponding software and hardware. The VR application was designed to allow the user to monitor the maintenance and development processes of the fossilite ferrous site on various timeframes and at different cartographic scales. This required the import of 15 photogrammetric models to the same project. The difficulties encountered were related to the management of a large number of 3D models as well as the maintenance of their photorealistic texture. Therefore, the area rendered in 3D was carefully selected by only cropping the area of interest and removing the faces of the model depicting the surrounding area. This procedure was applied to all 3D models used in the VR application. For the lighting of the scenes, natural directional lighting was created. The lights' direction was set at a 90° angle perpendicular to the rooms to eliminate shadows that affect the feeling of the space and the texture and colors of the 3D models. In addition, the rooms created using the ArcGIS Map SDK offered relatively limited interaction, as this plugin has been recently released and not all possibilities of connecting GIS to VR have been thoroughly researched.

Regarding the use of the VR application developed, some conclusions were drawn in terms of its functionality. It was initially observed that users who were not accustomed to VR equipment encountered difficulties in navigation. For example, users who are unfamiliar with VR hesitate with their movements in the virtual space. Furthermore, the simulation of the movements and interactions through the controllers was initially incomprehensible, as novice users were not accustomed to the use of this VR equipment. For this reason, it is recommended that users be instructed on the operation of VR equipment (headset and controllers) prior to the virtual tour. Practicing with the equipment does not require a lot of time, as it takes users just a few minutes to familiarize themselves with the use of the mask and the virtual space and to then learn how the actions and buttons operate in this application. The total training time ranges from 10 to 20 min, depending on the user's previous experiences with VR applications.

## 7. Conclusions

The main conclusion that can be drawn from the proposed methodology and the derivatives of the present study is that cartographic scale and data acquisition with UASs are crucial for the efficient visualization of geospatial information in VR. The cartographic derivatives from UAS data can be successfully geovisualized at the proper scale in a VR environment. Regarding the multitemporal monitoring of the maintenance and figuration of the fossilite ferrous site, VR geovisualization has many advantages such as: (i) the simultaneous observation of the fossil site at different times, (ii) the full utilization of the high resolution in the cartographic results, and (iii) the interactive conveyance of the deduced information.

The broad contribution of the present research is that the flight characteristics (flight altitude, flight path, and angle of the camera) for acquiring VHRIs strongly influence the quality of derived 2D maps and 3D models. Three mapping scales were required for high-resolution 2D and 3D mapping and the VR geovisualization of the fossilite ferrous site. The number of scales is associated with the the levels at which the spatiotemporal monitoring of the site's formation process were performed: (i) the fossil site (1:120), (ii) the fossil root system (1:20), and (iii) individual fossils ($\geq$1:10). Therefore, for their accurate and precise 3D mapping and their efficient (optimal) 3D VR geovisualization, the observance of cartographic principles and scale issues is of key importance. Therefore, the method of UAS data collection and the spatial resolution of VHRIs are determined by the cartographic scale. Overall, our results demonstrate a strong cartographic scale effect in the development of VR geovisualization and confirm that the use of 2D and 3D maps in VR offers new possibilities in the field of cartography.

To evaluate the usability of VR 3D mapping, a future goal is the conducting of further game-crowdsourcing experiences. The participants will appraise the qualitative and quantitative aspects of the application. Their recommendation will determine the advantages and disadvantages regarding the immersive experience. Future research will continue to explore the utilization of 2D and 3D maps in VR. More specifically, one of our future goals is to develop a protocol for UAS data acquisition in order to create high resolution 2D and 3D maps in VR. Furthermore, another research subject to be investigated is the transfer of thematic information and its efficient visualization on future 2D and 3D maps, as well as the combination of these two cartographic results in VR.

**Author Contributions:** Conceptualization, E.-E.P., A.P., N.-A.K., N.Z. and N.S.; methodology, E.-E.P., N.-A.K. and N.S.; software, E.-E.P.; writing—original draft preparation, E.-E.P., A.P., N.-A.K. and N.S.; writing—review and editing, E.-E.P., A.P., N.-A.K., N.Z. and N.S. All authors have read and agreed to the published version of the manuscript.

**Funding:** This research was funded by the Research e-Infrastructure "Interregional Digital Transformation for Culture and Tourism in Aegean Archipelagos" {Code Number MIS 5047046} which is implemented within the framework of the "Regional Excellence" Action of the Operational Program "Competitiveness, Entrepreneurship and Innovation". The action was co-funded by the European Regional Development Fund (ERDF) and the Greek State [Partnership Agreement 2014–2020].

**Data Availability Statement:** Not applicable.

**Acknowledgments:** We thank the editor and the three anonymous reviewers for their insightful comments which substantially improved the manuscript. We also thank Vlasios Kasapakis for his help with VR programming and Aikaterini Rippi for her help with English language editing.

**Conflicts of Interest:** The authors declare no conflict of interest.

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
