# Peer review of "VR Multiscale Geovisualization Based on UAS Multitemporal Data: The Case of Geological Monuments"

_remotesensing, doi:10.3390/rs14174259_

Round 1

Reviewer 1 Report

The paper explains in great detail the instrumentation and procedure that was use to develop a VR immersive system for the multi-temporal and multi-scale monitoring of a test UNESCO site. 

The content of the article can be interesting even for a non-scientific reader indeed, and there is no lack of detail in the description of the whole process that brought from the very first step of the UAV flights management and photos collection, to geoprocessing and mosaicing, and finally to the design of the rooms of Virtual Reality for the end-users.

I suggest to have another read of the article altogether, and make it more concise, less repetitive.

Minor editorial comments:

l.11: specify UAS acronym
l.50: can have a thus? missing word

Author Response

Dear reviewer. Thank you for taking the time to consider our manuscript. We are grateful for the constructive feedback. Considering the comments, we have carefully made the revisions to incorporate your recommendations. We also elucidate more on the aspects that were not clear in the previous manuscript version. We feel these changes make our manuscript appropriate for the readership of the Remote Sensing Journal.

Our answers and comments are the following.

The paper explains in great detail the instrumentation and procedure that was use to develop a VR immersive system for the multi-temporal and multi-scale monitoring of a test UNESCO site. 

The content of the article can be interesting even for a non-scientific reader indeed, and there is no lack of detail in the description of the whole process that brought from the very first step of the UAV flights management and photos collection, to geoprocessing and mosaicing, and finally to the design of the rooms of Virtual Reality for the end-users.

I suggest to have another read of the article altogether, and make it more concise, less repetitive.

Overall, the article was revised and made more concise.

Minor editorial comments:

l.11: specify UAS acronym

Done.

l.50: can have a thus? missing word

Corrected.

Reviewer 2 Report

This research paper focuses on VR multiscale geovisualization, which is a very interesting and popular topic. This paper investigates the correlation between UAS data acquisition flight characteristics and the quality of the derived maps and 3D models of geological monuments for Virtual Reality (VR)  geovisualization in different scales and timeframes. The English writing is fine.

In general, the contents are complete and the experimental results are valid. However, I think there are still a few steps the authors need to improve to publish this paper.

1 The major experimental results are qualitative instead of quantitative. I am afraid that it is not enough for a scientific research.

2 To better elaborate the advantages of your proposed technique, the authors need to present the traditional methods and make more detailed comparative experiments.

3 The geological monuments in VR achieves impressing effects; therefore, what are the main problems hindering its generalization in other different scenarios?

4 The authors should state the main contributions of this work. Although many details are presented and the intermediate processes are integrated, It is crucial to emphasize what is the difference between this work and similar previous work.

Author Response

Dear reviewer. Thank you for taking the time to consider our manuscript. We are grateful for the constructive feedback. Considering the comments, we have carefully made the revisions to incorporate your recommendations. We also elucidate more on the aspects that were not clear in the previous manuscript version. We feel these changes make our manuscript appropriate for the readership of the Remote Sensing Journal.

You will find our answers and comments in the attatched pdf.

Reviewer 3 Report

Excellent manuscript and very well written. Only several areas where wording might be improved upon. Many of the comments/questions I initially posted in the introduction and beginning of the methods were addressed, so please know that some comments were written before seeing that the information was later supplied. The attached PDF contains the comments and editing suggestions.

Author Response

(The authors gave the same response as above.)

Reviewer 4 Report

I think that the article does not bring new scientific opinions or procedures. It is rather a popularization article. The article needs to be fundamentally revised. It is necessary to add more details to the article about the measurement methodology, the accuracy, the processing procedure, etc.. it would be appropriate to make a detailed analysis of errors, compare the results with other articles, etc....

Author Response

(The authors gave the same response as above.)

Reviewer 5 Report

Dear Authors,

your paper describes the operations conducted by your research team for multitemporal 3D geovisualization of geological heritage on different cartographic scales by integrating UAS technology with VR techniques.  The paper is well set up and well written, but it feels like a topographical activity report conducted in archaelogical area. In my opinion the operations described are widely standardized and already used by the geomatics community and I don't detect any scientific innovation.

Author Response

(The authors gave the same response as above.)

Round 2

Reviewer 2 Report

The results of comparison experiments can be extended.

Apart from that, I do not have other comments.

Author Response

Dear reviewer. Thank you for taking the time to consider our manuscript. We are grateful for the constructive feedback. Considering the comments, we have extended the results of comparison experiments.

Reviewer 4 Report

Significant changes have been made to the article. The quality of the article has improved. The article presents the issue better, and the Scientific Soundness is also better.

Author Response

Dear reviewer. Thank you for taking the time to consider our manuscript. We are grateful for the constructive feedback.